# Participatory Design of Sonification Development for Learning about Molecular Structures in Virtual Reality

**Miguel Garcia-Ruiz** [1,*] **, Pedro Cesar Santana-Mancilla** [2] **, Laura Sanely Gaytan-Lugo** [2] **and Adriana Iniguez-Carrillo** [3]

1  School of Computer Science and Technology, Algoma University, Sault Ste. Marie, ON P6A 2G4, Canada
2  Facultad de Telemática, Facultad de Ingeniería Mecánica y Eléctrica, Universidad de Colima, Colima 28040, Mexico
3  Departamento en Ciencias Computacionales e Innovación Tecnológica, Universidad de Guadalajara, Guadalajara 44100, Mexico
*  Correspondence: miguel.garcia@algomau.ca

**Abstract:** Background: Chemistry and biology students often have difficulty understanding molecular structures. Sonification (the rendition of data into non-speech sounds that convey information) can be used to support molecular understanding by complementing scientific visualization. A proper sonification design is important for its effective educational use. This paper describes a participatory design (PD) approach to designing and developing the sonification of a molecular structure model to be used in an educational setting. Methods: Biology, music, and computer science students and specialists designed a sonification of a model of an insulin molecule, following Spinuzzi's PD methodology and involving evolutionary prototyping. The sonification was developed using open-source software tools used in digital music composition. Results and Conclusions: We tested our sonification played on a virtual reality headset with 15 computer science students. Questionnaire and observational results showed that multidisciplinary PD was useful and effective for developing an educational scientific sonification. PD allowed for speeding up and improving our sonification design and development. Making a usable (effective, efficient, and pleasant to use) sonification of molecular information requires the multidisciplinary participation of people with music, computer science, and molecular biology backgrounds.

**Keywords:** participatory design; molecular; model; sonification; virtual reality

## 1. Introduction

Chemistry and biology students often have difficulty understanding molecular information. Studies performed by [1] have shown that many students do not easily understand molecular structures due to their intricate three-dimensional structure and other physicochemical properties [2–7]. Traditional teaching tools (e.g., blackboard or plastic molecular models) have been used to teach general concepts about the structure of molecules, but they have some limitations in terms of facilitating the understanding of very specific and precise properties related to the 3D structure of molecules [8] that can be better perceived using computerized interactive models [9], with some limitations [10]. Past research reports that mapping sounds to data, also known as sonification [11], has been successfully researched and employed to support the study of molecular information by facilitating the comprehension of relationships of molecular properties, which are difficult to perceive through visualization alone [12].

Auditory Display studies the use of sound to convey meaningful information [13]. It has been researched and developed for some decades covering different application domains, taking advantage of the human sense of hearing, including its great capacities for uncovering patterns in sound and perceiving temporal changes [14]. In addition, Auditory Display can be useful for people with visual impairment, working as an alternative to

visualization. Auditory Display is an umbrella term that includes techniques and subsets for representing information with sound. Sonification, a subset of Auditory Display, utilizes non-speech sound properties to convey useful information to the human listener and to perceptualize data, usually generated by scientific research [15,16]. In the sonification of scientific information, data relationships can be easier to hear than to see [13,17]. Parameter-mapping sonification maps large (and often complex) datasets to specific audio properties such as pitch and loudness. Non-speech sounds from sonifications can be composed of musical notes made by digitally recorded or synthesized musical instruments and by pure tones such as sine or squared audio tones [18]. In addition, melodic and rhythmic sonifications can be used as a mnemonic device for supporting the teaching and learning of complex concepts [13], leveraging the human ear's capability of memorizing information patterns, and individual learning styles [19]. However, sonifications working as mnemonics must be correctly designed and tested to be effective [20].

There are other techniques used in Auditory Display called earcons and auditory icons. Earcons are synthetic and abstract sounds composed of musical notes that are structurally combined to represent information [21], for example, a group (also called motif or motive) of three musical piano notes representing an incoming email. Earcons can produce a pleasant and melodic auditory message, but the user must learn its abstract meaning at the computer interface [19]. Non-speech sounds from Auditory Display can also be composed of everyday sound effects that represent information or actions, called auditory icons [15,22], where there is a natural and easy-to-understand mapping between a non-speech sound and an action at the computer interface [17]. However, auditory icons and earcons have been rarely used in scientific sonification because they are mostly applied to providing direct feedback, mainly working as alarms and warning messages [18], although they can be used for representing molecular events such as confirming positive molecular docking using molecular models or representing molecular bonds with earcons or auditory icons [23]. Auditory Display techniques such as sonification using musical notes and earcons require previous training where users learn the mappings between the sounds and the information being sonified. This training must be carefully planned and considered when designing an Auditory Display application in the sciences. Spatial audio and sonification in virtual reality (VR) have been recently researched due to computer hardware and software advancements, enhancing user immersion and positional sound perception [24]. Sonification in VR in turn supports the comprehension of three-dimensional molecular structure models displayed in VR by offloading information from the human visual channel, as well as making visual information more salient in a 3D virtual environment [25–27], among other applications.

A very important (and often difficult) step of scientific sonification development is to properly design and test the mapping of non-speech sounds to scientific information, taking into account a number of sound properties such as timbre (it defines the type of instrument played), pitch (the perceived frequency of sound), tempo (the speed at which music is played), loudness (it defines the sound intensity), and the sound's spatial position, among other parameters. Using multiple sound parameters in sonification can be useful for representing multidimensional data displays [18]. Sonification can be developed by students and experts from different disciplines who can collaborate in the development of sonifications, such as biology, chemistry, molecular biology, computer science, music, etc. This is the basis of participatory design (PD), which is a design approach that fully involves an active and iterative participation of end users and product development stakeholders (which become co-designers) during the entire design, development, and testing process of a product or service, focusing on the user's orientation and collaboration [28,29]. One of the goals of PD is to improve understanding of the users' needs for designing the product or service. In our case, these needs are based on the learning objectives from the molecular model used in molecular biology and related courses, and the needs of molecular biology and chemistry students who will learn them. PD is an important approach applied to product design because potential end users of the product perceive a sense of ownership and acceptance of it [28].

The main objective of this paper is to describe the overall analysis, benefits, and challenges of PD applied in the molecular sonification development process. More specifically, this paper describes the multidisciplinary design of non-speech (musical) sounds mapped to molecular information, along with the 3D visualization of its respective molecular model. The paper also describes questionnaire and observational study results for assessing the degree of involvement of participants and PD in sonification's development. It concludes with a discussion on the usefulness of PD and lessons learned from our molecular sonification design process.

Our research question that we answer in this paper is as follows: Can participatory design (PD) effectively supports the design of non-speech sounds to be used in scientific sonification for supporting learning about molecular structures?

*Background*

The literature reports many research projects and applications over the past three decades concerning sonification utilized for analyzing molecular properties. The following are representative examples of those projects and applications. PROMUSE is a visualization/sonification system developed by [12] intended to analyze protein structure alignments in 3D, where some molecular data were represented aurally using melodic components. Researchers found that sound can help chemists discern and clarify molecular information, overcoming problems such as the visual occlusion of molecular structures. Ref. [30] made a detailed sonification of protein sequences from two influenza viruses using the audification technique, where protein data are directly translated to the auditory domain by mapping the audio tone duration and frequency, helping to produce a distinctive "signature" for the biological functions of those proteins. Ref. [25] describe the development of a large visualization system that displayed molecular structures using 3D stereoscopic visualization along with its sonification. The researchers found that the sonification of molecular properties enhanced users' immersion and complemented and enhanced the visualization of interactive molecular simulations, allowing users to focus on events extracted from a molecular dynamics simulation. Ref. [31] developed a sonification of deoxyribonucleic acid (DNA) molecules consisting of six sonification algorithms that mapped sequential groups of nucleotide bases (DNA motifs). A proof of concept revealed that some DNA sequence properties can be identified through sonification alone.

Sonifications carried out in personal computers have been proven to support the comprehension of molecular behavior, since sonifications can make abstract information more concrete and salient and can complement scientific visualization in multimodal human–computer interfaces, among other educational benefits [15]. Moreover, sonification can improve students' motivation and understanding of abstract information [32], not to mention the invaluable support of sonifications for visually impaired students as an educational tool. A thorough overview of molecular sonification for learning is described in [33]. What follows describes a sample of relevant research on auditory display applications for the analysis and study of molecular information. Ref. [23] conducted a study on the sonification of molecules using earcons (musical sounds) that represented data for amino acids in a 3D computer visualization system, which were otherwise difficult to analyze if graphical models were solely visualized. Results of this research showed that amino acid sonification, in combination with visualization, is useful for conveying and recalling their molecular properties and structures. Ref. [34] developed an audio-visual computer program (a browser) intended for analyzing structures of ribonucleic acid (RNA) molecules, developing novel audiovisual representations of RNA shapes, adding extra sensory information to support search tasks within the RNA molecule, and facilitating comprehension of RNA shape information. Ref. [35] taught sonifications of DNA sequences to high school students, with compelling results related to students' motivation.

The design of molecular sonifications should be carried out by people from different disciplines and backgrounds due to its technical (and sometimes artistic) and multidisciplinary nature, following a design philosophy such as participatory design (PD). It is

a mature research field that started in the Scandinavian countries in the 1970s and has been applied in the design of interactive systems and human–computer interfaces ever since [28]. PD has been considered as a powerful design approach where future end users of an interactive system/computer application continuously collaborate by participating as co-designers in the system's design process. This design process may also involve other types of participants in some co-designing tasks, such as stakeholders involved on the software/hardware development team who belong to disciplines other than design (e.g., marketing, ergonomics, etc.). PD generally includes persons with different backgrounds, interests, and areas of expertise. However, one of the main challenges in PD is the adequate coordination of all multidisciplinary people in PD activities [36]. PD is a user-centered design approach because it commits to essential and non-essential end users' needs and desires that will shape the design of the interactive system, considering designers and end users' attitudes and mindsets [29]. In PD, end users should engage in the design process, being aware of possible design problems and proposing valuable initial feedback during early design discussions and stages [37,38]. PD has been recently applied in the development of educational virtual reality systems. Ref. [39] reports a PD-based methodology where students are central in educational VR design, allowing them to participate actively in its development. PD can also be applied to developing educational VR for science teaching. Ref. [40] created an educational VR environment for teaching about ocean acidification, developed in conjunction with marine educators. By participating in development of educational VR, marine educators found that 3D visualization, empowerment, and perspective taking may support teaching about ocean acidification. Thus, educators become designers of new learning environments [41].

PD has also been researched and applied in the development of data and user interface sonification, for example, in [42–44]. However, Ref. [45] found that sonification design is challenging because a number of factors, including the relatively recent research about it (it has been around for 25 years, compared to other data analysis techniques) and the difficulty of correctly designing the mapping between sound and data, and points out the importance of the participation of stakeholders such as researchers and end users in the sonification design. These challenges can be overcome using PD [42]. Despite its importance, the literature shows few papers describing the application of PD approaches for designing and developing scientific visualization or sonification systems, e.g., [44,46], which has potential for future research and development.

Sonification has been researched for over two decades for supporting the understanding of molecular structures and their properties. However, very few research papers describe how and to what extent molecular sonifications were developed through multidisciplinary/interdisciplinary collaboration and/or using PD or similar design approaches. For example, Ref. [47] points out the importance and value of collaboration among software developers, molecular biology specialists, and end users in the molecular sonification development process, and the importance of the early involvement of users in the design process of their molecular docking system. Ref. [47] also emphasized the need for continuous end-user participation in the development process and for having molecular docking experts evaluate their molecular sonification prototype. However, this study does not deal with the sonification of molecular properties such as hydrophobicity, essential for analyzing molecule–molecule interactions, and it does not explain whether molecule–sound mapping was designed by the end users. Ref. [48] developed a computer system that reads out loud the names of chemical components, including biological molecular models, to visually impaired students. Ref. [48] reported that the system was collaboratively developed by an audio engineer, two computer scientists and one chemistry teacher and researcher, and five chemistry students, and was tested by eight people, including three totally blind students. Testing results show that testers correctly identified 90% of the molecular components. The work of [49] describes the multidisciplinary collaboration between a biologist and an artist in the sonification of protein data. This article goes on to explain how the collaboration between art and science was completed, sharing concepts such as patterning and

aesthetics. Ref. [50] describe an interdisciplinary project on the design and development of the sonification of molecular data related to Amyotrophic Lateral Sclerosis (ALS) disease. Its multimedia sonification was exhibited in the University of California at Los Angeles' Arts/Sci Gallery, with the objective of engaging the audience and raising awareness of ALS. The literature reports other molecular sonification developments, but it is unclear whether their authors applied PD or similar approaches in their sonification design and development.

## 2. Materials and Methods

The literature reports molecular sonifications designed and made by chemistry and computer science researchers which are intended for professional and research use, e.g., for conducting structural protein research. In addition, some of those sonifications have been created and researched for educational purposes. However, to our knowledge, there are no reported molecular sonifications designed and implemented with the active and continuous participation of end users such as chemists or students following a structured and systematic participatory design approach. We are interested in applying PD in the design, development, and testing of molecular sonification, completed by students from multiple disciplines. We invited music, chemistry, biology, and computer science undergraduate students in the design and making of our sonifications. Our sonification application is designed on a participatory design (PD) paradigm where we designed with students in mind, leveraging their active participation in the development of sonification [29]. The students collaborated continuously in the design of our sonification, guided by computer science and molecular biology experts.

Thus, our development team consisted of:

- Two students with both a music and computer science background. This was very helpful because they provided excellent musical feedback based on digital musical instruments;
- a music student;
- two biology students;
- three computer science researchers who are experts in usability and human–computer interaction (HCI).

Figure 1 shows the students who collaborated in the sonification design.

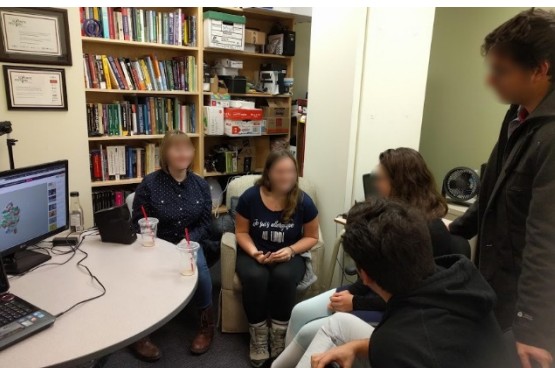

**Figure 1.** Students discussing the molecular sonification design.

We followed Spinuzzi's PD methodology [51], based on these three iterative stages:

(1) Initial work exploration: Experts meet with students and expose them to molecular models and sonification examples and let them explore software tools. In this stage, we also include a concise introduction on participatory design to educate all participants about it and to avoid user involvement obstacles, described by [52].

(2) Discovery processes: experts and students participate in focus groups focusing on designing sonifications and describing their meaning.

(3)   Prototyping: Experts and students jointly and collaboratively develop the sonifications.

The three stages are iterative and are described in more detail in Figure 2. The Kanban board was updated in each of the stages according to the stakeholders' progress and activities in the project.

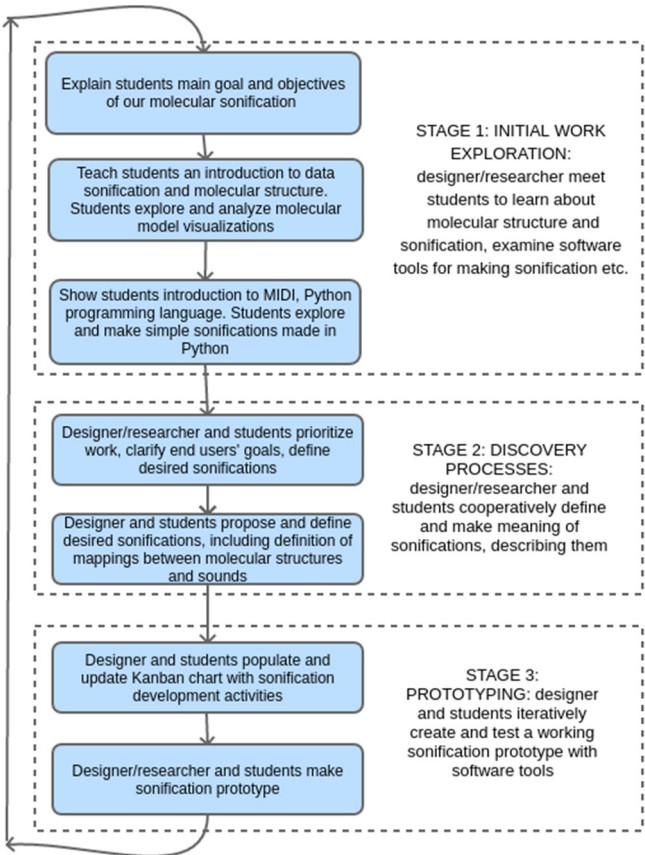

**Figure 2.** Our PD methodology, based on Spinuzzi [51].

We emphasize in the sonification process the importance of multidisciplinary work, where students contribute actively in their knowledge fields (e.g., biology) in our participatory design process. In it, undergraduate computer science and biology students participate in the design of the molecular sonification. In addition, we invited undergraduate music students to participate in the sonification development. First, we showed the capabilities of our software tools and code we developed to the students. We explained to the students what data sonification is and its benefits and challenges. We also explain them what is possible to sonificate and what kind of MIDI's synthesized musical instruments, and how their properties can be used in our sonifications with our programs. Second, we developed a medium-fidelity prototype with the VR molecular display and its sonification, based on sonification development guidelines described in [11]. Third, each student tried the VR molecular sonification by donning a VR headset. Then, we ran a focus group with the students to analyze our sonification prototype(s), discuss sound mappings, and suggest improvements/new sonifications. The first author moderated the focus group. The activities conducted in our development stages are in line with [53]'s research, where they applied PD based on focus groups, collaborative and iterative prototyping, and qualitative testing for the development of sonifications.

It was very useful to create a Kanban board (also called Kanban chart). A Kanban board is a workflow project management tool that originated in Japan, allowing for all the stakeholders' project tasks to be communicated and kept track of [54]. The Kanban

board can have three or more columns showing the project tasks' status, such as To Do, In Progress, Testing, and Done. In physical Kanban boards, the task names are generally written on sticky notes that are attached to each column. The Kanban board is updated by adding new sticky notes to it and by moving the sticky notes from one column to another according to each person's task completion. It is also possible to use an online Kanban tool such as Trello [55], which is the one that we successfully used in our project.

### 2.1. Sonification Development

The first author of this paper wrote a program in Python language that generates musical notes to make the sonification of molecular structures, running on Ubuntu Linux. We used a Python library called BioPython [56] to open and parse a Protein Data Bank (PDB) file, a standard file format that contains three-dimensional structural information of biological molecules such as proteins. We also applied BioPython to identify the molecule's amino acids (the building blocks of proteins). Our program also used the MIDIutil library [57] to assign (map) amino acids to musical notes by saving them to a Musical Instrument Digital Interface (MIDI) protocol format file. This file was then converted to MP3 audio file using the software-based MIDI sound renderer Timidity [58]. We created a graphical molecular model based on the PDB file using the molecular visualization software tool VMD [59]. The generated MP3 file, along with the 3D molecular model, were uploaded to a website called Sketchfab [60]. The molecular model can be visualized and the sonification can be heard on the Sketchfab website from any personal computer and operating system that handles WebGL. Figure 3 summarizes the steps that we followed for creating the molecular sonification.

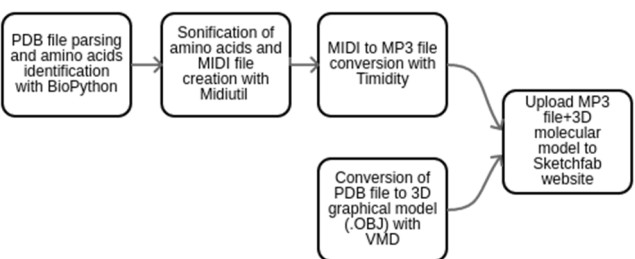

**Figure 3.** The sonification development process.

The Sketchfab app, when opened on a smartphone, allows for visualizing 3D graphical models in virtual reality mode (stereoscopic display). The smartphone was then inserted into a virtual reality (VR) headset to provide an enhanced immersive experience. People upload 3D models and the accompanying sound to Sketchfab, forming an online catalog. Sketchfab uses the phone's gyroscope (a type of sensor that perceives the phone's position and orientation in 3D) to change the molecular model's point of view according to the head's movements (shown in Figure 4). Our sonification is not generated interactively on the website; rather, we wanted to create a proof of concept with the sonificated molecular model and conduct initial user tests with students about the usability of a pre-rendered molecule sonification in conjunction with its respective molecular model's visualization. Usability measures how efficient, effective, and pleasurable a software-based user interface is [61,62].

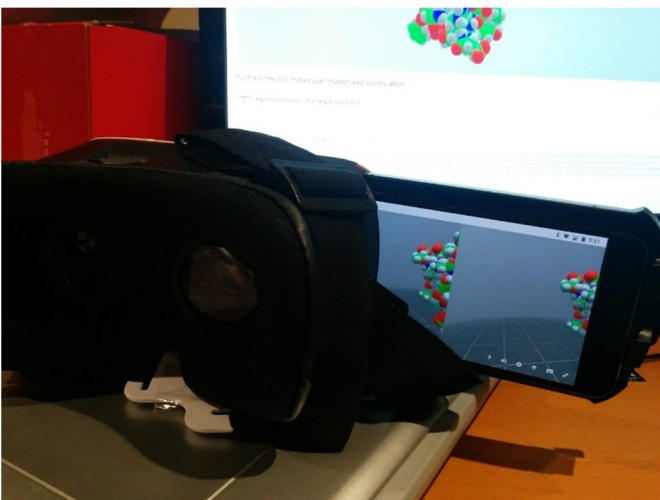

**Figure 4.** A VR headset displaying a molecular model using Sketchfab.

We used a molecular model of insulin (a hormone that the human body uses for controlling blood glucose levels) as a testbed for our first sonification prototype, which is shown in Figure 5. Table 1 shows mappings between musical notes from our sonification (working as earcons) and all 20 basic amino acids that make up molecules such as the insulin molecule. This sonification was initially played, and the molecular model was shown to the students who collaborated in the project to explain them the topic of sonification and to get ideas on further sonifications. We mapped each amino acid's hydrophobicity type from the insulin molecule to musical notes. The hydrophobicity of molecules is a physical property that determines important intermolecular interactions, particularly how soluble an amino acid is in water. It also determines the position of the amino acid in a larger molecule. For example, very hydrophobic amino acids are likely to be found in the interior of the molecule [5].

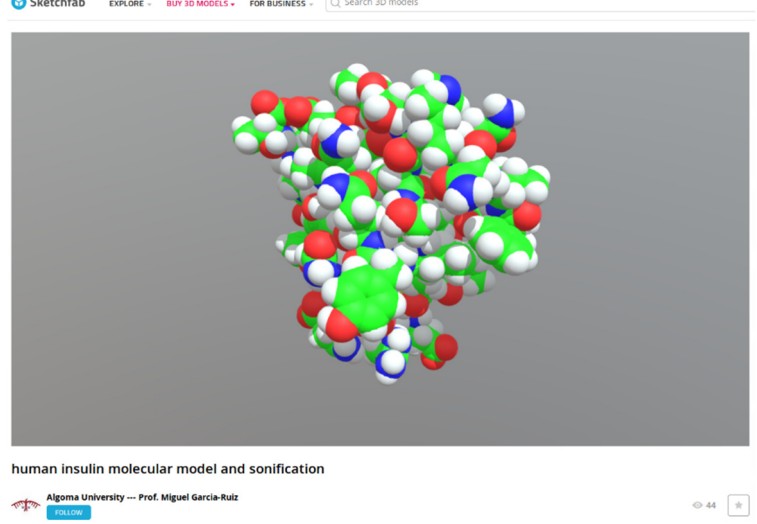

**Figure 5.** The molecular model of insulin shown on the Sketchfab website.

The main author mapped four musical instruments' hydrophobicity types using MIDI format:

- Very hydrophobic (the amino acid is very repellent to water) = acoustic grand piano.
- Hydrophobic (the amino acid is repellent to water) = violin.
- Neutral = trumpet.
- Hydrophilic (the amino acid is attracted to water) = xylophone.

The authors selected those four musical instruments because they wanted them to sound distinguishable and rather pleasant in our sonification. The sounds were used to make an initial prototype, working as a testbed. The main author then pre-recorded the sonification using the MIDI file format. Each of the basic amino acids' musical notes were played with a slight duration and pitch difference to make the sonification more "musical", and this may help in the identification of each amino acid. We followed the earcon design guidelines developed by [63]. The purpose of our sonification was to make the hydrophobicity of the insulin's amino acids noticeable and easy to remember. New musical instruments and mappings will be added to the sonification repertoire in future prototypes according to end users and project participants' suggestions.

**Table 1.** Mappings among amino acids' hydrophobicity [64] and musical instruments.

| Amino Acid Name | Hydrophobicity Type | Musical Note's Instrument |
| --- | --- | --- |
| Glycine | neutral | trumpet |
| Proline | hydrophilic | xylophone |
| Methionine | very hydrophobic | acoustic grand piano |
| Valine (Val) | very hydrophobic | acoustic grand piano |
| Leucine (Leu) | very hydrophobic | acoustic grand piano |
| Isoleucine (Ile) | very hydrophobic | acoustic grand piano |
| Alanine (Ala) | hydrophobic | violin |
| Cysteine (Cys) | hydrophobic | violin |
| Arginine (Arg) | hydrophilic | xylophone |
| Lysine (Lys) | hydrophilic | xylophone |
| Threonine (Thr) | neutral | trumpet |
| Phenylalanine (Phe) | very hydrophobic | acoustic grand piano |
| Tryptophan (Trp) | very hydrophobic | acoustic grand piano |
| Glycine (Gly) | neutral | trumpet |
| Serine (Ser) | neutral | trumpet |
| Glutamic acid (Glu) | neutral | trumpet |
| Tyrosine (Tyr) | hydrophobic | violin |
| Histidine (His) | hydrophilic | xylophone |
| Aspartic acid (Asp) | neutral | trumpet |
| Asparagine (Asn) | hydrophilic | xylophone |

The insulin molecular model and its respective sonification can be freely accessed from the Sketchfab website [65]. The molecular model can also be found on the Sketchfab app by typing the keywords "Algoma University" in the app's search field. We chose the insulin molecule because of its importance in medicine, and its structure is small but complex enough for educational purposes [66].

A video abstract that illustrates the use of the sonification in the VR environment running on Sketchfab is found in: http://people.algomau.ca/garcia/PD_video.mp4 (accessed on 11 August 2022).

## 2.2. Tasks Completed by the Project Participants

Following Spinuzzi's PD methodology [51], the project participants selected and opened the molecular model of insulin from the Sketchfab website used for the sonification. The computer science students and the expert conducted software testing of the developed system, particularly conducting white-box testing (analyzing and testing the code modules

used for developing the sonification) and black-box testing (usability testing), ensuring the software's quality and effectiveness. The arts students focused on the musical properties of the sonifications, such as the melody, pitch, and timbre. At the end of the sonification development, all the students and specialists conducted a usability test of the sonifications played in an online 3D visualization site. The Kanban board was updated accordingly in each phase from our methodology. The molecular model shown on the Sketchfab website was displayed on a 24″ computer monitor. In addition, the project participants tested both the sonification and the 3D molecular visualization of the insulin molecule on a Google Pixel 2 smartphone, which was inserted on an EVO VR MI-VRH03-199 virtual reality headset (shown in Figure 4).

**3. Results**

In order to improve our sonification design process and its application, we wanted to see to what extent student participants were involved in the sonification's design and development. Previous research indicates that positive user involvement in PD generally leads to developing software systems with high usability [52,67]. To evaluate the degree of participants' involvement in the sonification design, they filled out a questionnaire based on [68]'s research on end-user computing development involvement. The following shows a summary of the participants' answers:

1. How satisfied are you with the application?
   *Not satisfied at all (1) … Very satisfied (5): _4__*
2. What aspects of the application, if any, are you most satisfied with and why?
   *Combining visuals, music, and science to enhance learning. The variety of musical instruments used in the sonification.*
3. What aspects of the application, if any, are you most dissatisfied with and why?
   *The musical sounds (sonification) are hard to understand without explanation.*
4. If applicable, please describe your involvement in the development of this application.
   *Providing input on the application.*
5. To what extent were you involved?
   *Suggestions were offered on sound design, including the type of musical instruments and their notes' structure.*
6. Would you have liked to have been more involved? If yes, in what aspects?
   *Students wanted to continue participating in the sonification design and development. Some students were interested in the coding aspect.*

According to [68], we also evaluated actual (how long a person spent participating in development activities) versus desired (how long a person wanted to spend in development activities) user involvement in our sonification development process. To do this, we adapted some of [68]'s questions (shown in Table 2), where each question has a Likert-type scale (not at all = 1; a little = 2; moderately = 3; much = 4; a great deal = 5). Table 2 shows the averages of the questionnaire results, answered by the project participants, marked in bold.

Table 2 indicates that although the five participants felt like they were indeed involved in the sonification design, they wanted to participate more in the project in aspects such as finding information sources (e.g., academic articles regarding molecular sonification) and coding and making the actual musical sounds.

Figure 6 shows the Likert-scale results in a graph form. The differences in scale results from question 4 are noticeable there.

**Table 2.** Questions asked to participants to analyze user involvement in the sonification design and their answers.

| Questions | Please Circle the Response That Best Describes Both Your Actual and Your Desired Participation in Each Activity. |
|---|---|
| | **Likert Scales** |
| 1. Initiating the sonification project? | ACTUAL: 1 2 **3** 4 5<br>DESIRED: 1 2 3 **4** 5 |
| 2. Determining the end user's information needs necessary for designing the sonifications? | ACTUAL: 1 2 **3** 4 5<br>DESIRED: 1 2 3 **4** 5 |
| 3. Assessing alternative ways of meeting the end user's information needs? | ACTUAL: 1 2 **3** 4 5<br>DESIRED: 1 2 **3** 4 5 |
| 4. Identifying sources of information (on how to properly design the sonification)? | ACTUAL: 1 2 **3** 4 5<br>DESIRED: 1 2 3 4 **5** |
| 5. Developing (making) the sonifications? | ACTUAL: 1 2 **3** 4 5<br>DESIRED: 1 2 3 **4** 5 |

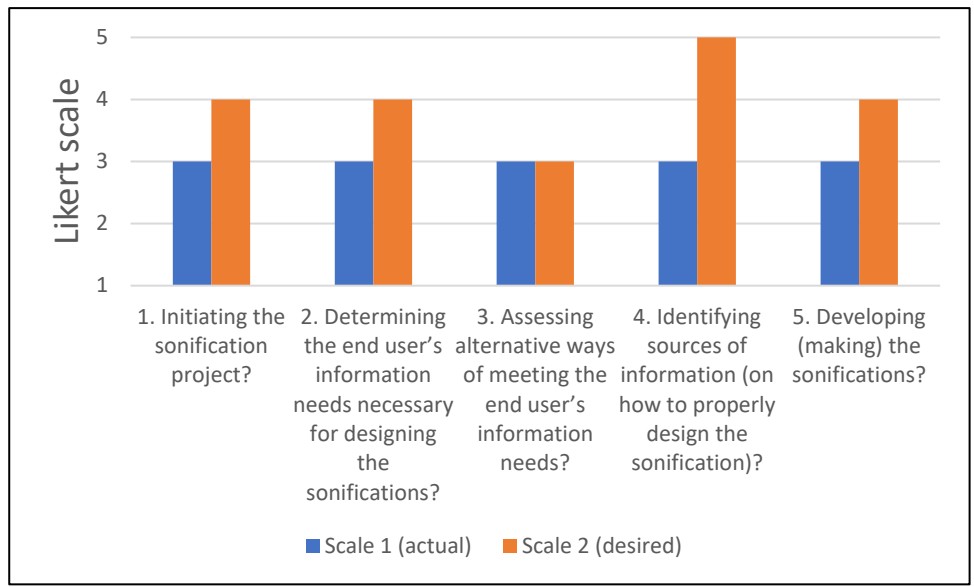

**Figure 6.** Graph showing the Likert-scale results from Table 2.

Figure A1 in Appendix A shows the scatter plots for all the Likert-scale data from the five questions. The scatter plots indicate some variability in the Likert-scale answers. The number of participants is small ($n = 5$) for obtaining statistically significant results from the Likert scales. For the sake of analysis, we calculated the standard deviations of those results, presented in Table 3.

**Table 3.** Standard deviations for the Likert scales from each question.

| | Question 1 | Question 2 | Question 3 | Question 4 | Question 5 |
|---|---|---|---|---|---|
| Scale 1 (actual) | 0.70710678 | 0.83666003 | 0.4472136 | 0.4472136 | 0.54772256 |
| Scale 2 (desired) | 0.4472136 | 0.4472136 | 0.70710678 | 0.54772256 | 0.54772256 |

Table 3 shows that question number 2 has the highest standard deviation, where perhaps participants were more hesitant to determine the end user's information needed (e.g., what kind of musical sounds were needed) for designing the sonifications.

We carried out a qualitative study by running a focus group with the students participating in the sonification development, asking what kind of musical sounds could be used in our proposed molecular sonification system. The focus group coordinator (the first author) asked the students to focus on possible auditory parameters (e.g., timbre, tempo, musical scales, melody, loudness, pitch, length, etc.), as well as other musical features such as motifs and chords, to be used in the sonifications. The students with musical training were asked to explain the musical parameters and features to the biology students, which did not have musical training. This way, all the students knew their meaning before making comments on the sonification in the focus group. Table 4 shows participants' relevant comments on the musical sounds to be used in the sonification. Their qualitative feedback is valuable and will be considered for developing further sonification prototypes.

**Table 4.** Comments made by the participants suggesting new sonification mappings.

| Participant Type | Comments |
|---|---|
| Computer science student with musical background no. 1 | *"Make earcons as chords so they will sound more melodic"* <br> *"Why not use the following instruments for mapping the amino acid properties? Piano, guitar, trumpet, xylophone, recorder"* |
| Computer science student with musical background no. 2 | *"Use water flowing chords for representing hydrophilic amino acids"* <br> *"Change tempo to make a difference, e.g., some amino acids played at slower pace"* |
| Biology student no. 1 | *"Change pitch of notes so that will make them more noticeable"* |
| Biology student no. 2 | *"Use longer musical notes for longer molecular chains"* |
| Music student | *"Use same musical scale for everything"* <br> *"Use chords for each amino acid. Chords can suggest moods".* <br> *"Use motifs to help memorize amino acids' properties"* <br> *"Use real motifs from existing songs. Use 20 of them, one for each amino acid"* |

We conducted a thematic analysis [69,70] on the qualitative data obtained from the focus group (summarized in Table 4) that we ran with the student participants. We identified from the students' comments four main themes related to sound parameters that can be used for making molecular sonifications:

- Use melodic sounds
- Use natural sounds (sound effects)
- Change musical notes' pitch
- Change motifs' length (a motif is a short melodic piece containing a succession of musical notes)

According to the thematic analysis, some students suggested to slightly change the notes' tempo and pitch from the sonifications. All the students agreed that using five different musical instruments could be useful for representing the different types of hydrophobicity. This may help future users to discriminate between musical notes among all the sonifications.

It is interesting to note that the musical parameters and features mentioned by the students in the focus group were the following:

- melody
- pitch
- timbre
- chord
- length
- scale
- motif

Those musical parameters and features have been used in sonification elsewhere, e.g., [13,71].

It is also worth mentioning that results of the qualitative analysis also show that the students did not mention the sound parameter of loudness. Loudness is a subjective measure of how intensely human ears perceive a sound and can be used in three-dimensional positional sound simulation. This may be useful for identifying the position of the sonification of a molecular piece in a molecular structural model in virtual reality [25]. In addition, loudness has been also used in molecular sonification for identifying amino acids from molecular structural models [71].

The students and specialists' feedback served to make and improve a new set of earcons, shown in Table 5. They were used in a molecular visualization program called PyMOL™). A Python program played the earcons according to the user's selection on a molecular model of insulin. A usability test on the new earcons and the PyMOL interactions is reported in [72]. In a study run with computer science undergraduates, it was found that the earcons were useful for representing the hydrophobicity of amino acids and positively complemented the molecular visualization of the human insulin molecular model.

**Table 5.** Mappings of amino acids and earcons played in P™ (TM).

| Amino Acids | Hydrophobicity Type | Earcon's Musical Instrument | Musical Notes |
|---|---|---|---|
| Leu, Ile, Phe, Trp, Val, Met | very hydrophobic | vibraphone | F#5, G5, G#5 (ascending notes) |
| Cys, Tyr, Ala | hydrophobic | xylophone | D#5, E5, F5 (ascending notes) |
| Thr, Glu, Gly, Ser, Gln, Asp | neutral | acoustic grand piano | C4, C4, C4 (Equal notes) |
| Arg, Lys, Asn, His, Pro | hydrophilic | flute | G#5, G5, F#5 (descending notes) |

We converted the amino acid and earcons mappings from Table 5 into a unique MP3 sound file, which we uploaded to the Sketchfab website that displayed the insulin molecular model. To analyze the ease of use and learnability of the earcons shown in Table 5, we ran a usability study with 15 computer science students (12 men, 3 women), with ages averaging 20 years old. In a small-scale usability test such as the one we carried out, past research calculated that 15 participants could uncover up to 100% of the usability problems of a software application [73]. More participants will be needed in further studies to get conclusive results in terms of the educational gains of our developed sonification.

We first trained the participants by showing Table 5 to them and playing the earcons separately, with the objective of learning about the mappings. The main task in the test was to observe the molecular model of insulin by wearing the VR headset and listening to our earcon sonification through a pair of HD206 Sennheiser headphones. After listening to the sonification and observing the molecular model twice, the participants filled out the System Usability Scale (SUS) questionnaire [74,75]. The questionnaire results indicated that the molecular VR visualization and sonification was highly usable, averaging 79.3% of the SUS' usability score. A SUS score that is over 68 is generally considered above average system usability, and a score of 78.7 or more indicates a very positive usability [76]. Most of the students reported positive user satisfaction when listening to the sounds, and some of the students reported them to be "melodic". At the end of the testing, we asked the students if they could recall some of the mappings by asking about them. On average, students correctly recalled 50% of the amino acid mappings.

## 4. Discussion

It is important to note that Sketchfab recently disabled sound playing capabilities due to some legal reviews on how the company handled sound on the web browsers, although the company developed a workaround [77]. Our sonification development and testing that includes playing sound on Sketchfab was conducted before sound was disabled on the Sketchfab website.

We found out in the usability study conducted with the 15 students that the earcons may be useful for supporting learning the amino acids present in a complex molecule such as insulin, but further tests are needed to corroborate this. The usability of educational software is regarded as an important factor that influences knowledge acquisition [78], hence the importance of running usability studies with sonifications to be used in an educational setting. Students could correctly recall just half of the amino acid mappings with the sounds perhaps due to the large number of mappings that they needed to learn in a short time. Further tests will indicate whether students could recall more mappings if the students are allowed to learn them for a longer time and with more repeated practice. It seems that the earcons used in the mappings raised students' interest about learning the amino acids, which can be related to the SUS questionnaire results and students' verbal comments made after the test. Further tests will analyze students' intrinsic motivation and engagement when learning about molecular structures supported by sonifications. This is in line with past research completed on learning about the sciences with sonifications for augmenting visual learning materials, e.g., [33,79].

The following discusses lessons learned and recommendations for further sonification developments employing PD, based on the results obtained from our sonification development:

- Make sure to explain clearly and concisely to the project stakeholders what types of technologies (e.g., VR) are being used in the project, especially to non-computer-science people. Some students or specialists who are not versed in technology may have difficulties in understanding how sonification works. It is necessary to explain this at the beginning of the project.
- Explain to all the stakeholders all the possible sound options and parameters for making the sonification mappings. Explain everything so that non-musical participants can understand their meaning without overwhelming them with musical theory. This way, the stakeholders will have a much-informed idea on what can be used in the sonifications. For example, at the beginning of our project, the first author showed a list of the musical instruments that can be synthesized with MIDI using our Python program and played each one of them, to get them familiar with their sound repertoire. He also explained that some auditory parameters could also be used, such as changing the musical instrument's duration and pitch.
- Molecular model sonifications may work as an example for teachers from other knowledge areas. Our generated molecular sonification may inspire teachers from other disciplines for at least considering applying sonifications in their courses. It is possible to develop sonifications in other knowledge fields, for example, in mathematics or economics. In addition, there are online examples of sonifications from other knowledge areas that can be used in class. For example, there is a sonification about the decline in U.S. coal production over the past 30 years, found in [80], that may be used in an economics course.
- Participants should feel like they are in control of the sonification design, allowing them to make decisions on its design. This is in line with participatory design philosophy, where participants should be motivated and committed to the design process activities [44].
- Early end-user involvement (e.g., chemistry and biology students) in the sonification design is of paramount importance. They will provide valuable initial feedback on the types of sounds used in the sonification, their mapping, etc., and this, in turn,

should support the creation of a smooth, engaging, and usable molecular sonification system [47].

- Keep all the stakeholders motivated in all the meetings, sessions, and user tests. Let them know what goals have been accomplished and that their participation has been valuable for achieving those goals. In turn, this will keep them participating in the sonification design and development process.
- Conducting short but intensive meetings with all the stakeholders is a good way to let them focus on the sonification designs, and for carrying out brief but effective brainstorming sessions. Document all the short sessions! Take notes, and if possible, video record them, so all comments from the stakeholders are kept for further analysis.
- All the stakeholders should participate in the creation of an initial sonification prototype at the beginning of the project. Participants with arts and computer science backgrounds should get together and create brief musical notes and compositions together in an early session, working as simple prototypes for all the stakeholders to discuss and improve upon in subsequent sessions. The stakeholders should be presented with a range of different non-speech sounds that can be considered for making the mappings to the molecular information. Choosing the right sounds facilitate the memorization of the mappings.
- It is important to conduct user testing with all the stakeholders from the development team, where they test the sonifications and the VR environment. They should evaluate the molecular model's sonification and its VR application considering the students and experts' points of view and user experience (UX), which may be complementary and/or different from the points of view obtained from the target users (e.g., biology and chemistry students).
- The Kanban board is a useful tool for keeping track of all the sonification development activities, including the testing process. It allows the stakeholders to be aware of all the projects and each person's progress.

*Issues and Challenges of Molecular Sonification Development*

At the beginning of the project, an important issue was that most of the students who participated in the sonification development did not know the basics of molecular structures. The authors asked the two biology students and the biology expert to share web pages that clearly and concisely describe molecular structures and are about amino acids, as the music, arts, and computer science students had to learn about it from those web pages. The molecular information was also reviewed by the participants with a biology background in our meetings.

There are some issues and challenges associated with designing and developing sonifications in the sciences, and we needed to tackle them appropriately and timely when we apply PD. A major challenge in molecular sonification is to determine the right mapping between sound features and molecular data. As [13] pointed out, it is difficult to determine the right musical instrument to map to a specific piece of molecular information, since this mapping is not natural, and users will need to learn it. In addition, the mapping could be done either randomly, following an algorithm, or by manually assigning musical notes and instruments to molecular structures. One critical activity of our sonification process is learning the mapping between musical notes and molecular information, because it is precisely the mapping that future students will learn and understand once the sonification is ready to use to support teaching and learning about molecular structures.

## 5. Conclusions

This paper described the participatory design of a molecular model's sonification, where computer science, biology, and music students actively participated in its design, guided by usability and human–computer interaction experts. The most important findings were that multidisciplinary participation allowed for speeding up and improving our sonification design process. In addition, students learned how a small research project is

conducted, enriching the sonification development. The positive usability testing results provided important insight into the usefulness and ease of use of the sonifications. These results support the importance of HCI in the sonification development process.

Our findings answered our research question, confirming that the participatory design methodology effectively supports the design of non-speech sounds to be used in scientific sonifications used for learning about molecular structures. This is in line with the results obtained from the end-user computing development involvement questionnaire that we applied in our PD process.

The collaboration of HCI experts was also important since they provided students with valuable expertise and guidance in the project. This was a win–win situation: both students and professors learned about sonification and contributed to make our sonification a better learning tool. Thus, PD is a very useful and important approach that should be used in the design, development, and testing of educational technology tools. Student participation was crucial for reporting the system's usefulness and designing it according to the users' requirements and needs. The user involvement assessment confirmed the usefulness of PD in our sonification development.

We found that Python programming language is simple yet powerful enough for developing sonification prototypes. Python allowed us to rapidly integrate students' and professors' opinions into the sonification development. This programming language is easy to learn, allowing non-computer-science people get a grasp on its capabilities for performing sonifications. The use of open-source software tools for creating MIDI files was proven to produce a fast software prototype, and participants in the project provided useful feedback early in the project. Designing a molecular model's sonification with high usability (effective, efficient, and pleasant to use) definitively requires the multidisciplinary participation of people with music, computer science, and molecular biology backgrounds. This should, in turn, contribute to a positive user experience for the sonification.

Future work will include testing our designed sonifications with students in a classroom, displaying the molecular model using a data projector and playing the molecular sonification using a pair of speakers. We will also conduct sonifications of more complex molecules, allowing more time for students to learn and practice the mappings between the sonifications and the molecular models.

**Author Contributions:** Conceptualization, M.G.-R.; methodology, M.G.-R.; software, M.G.-R.; validation, M.G.-R., P.C.S.-M. and L.S.G.-L.; investigation, M.G.-R., P.C.S.-M., A.I.-C. and L.S.G.-L.; writing—original draft preparation, M.G.-R.; writing—review and editing, M.G.-R., P.C.S.-M., A.I.-C. and L.S.G.-L. All authors have read and agreed to the published version of the manuscript.

**Funding:** This research received no external funding.

**Institutional Review Board Statement:** The study was conducted in accordance with the Declaration of Helsinki and approved by Algoma University's Research Ethics Board (REB) for studies involving humans.

**Informed Consent Statement:** Informed consent was obtained from all subjects involved in the study.

**Data Availability Statement:** Data supporting reported usability testing results can be found at: http://people.algomau.ca/garcia/PD_sonification_SUS_questionnaire_results.xlsx. Accessed on 11 October 2022.

**Acknowledgments:** The first author acknowledges support from Algoma University's Interactive Gaming Technology Lab. This study was developed within the framework of the project "Use of low cost hardware and videogames in classes across México and Canada" from Algoma University and the Universidad de Colima. All the authors have read and agreed with the Acknowledgements.

**Conflicts of Interest:** The authors declare no conflict of interest.

**Appendix A**

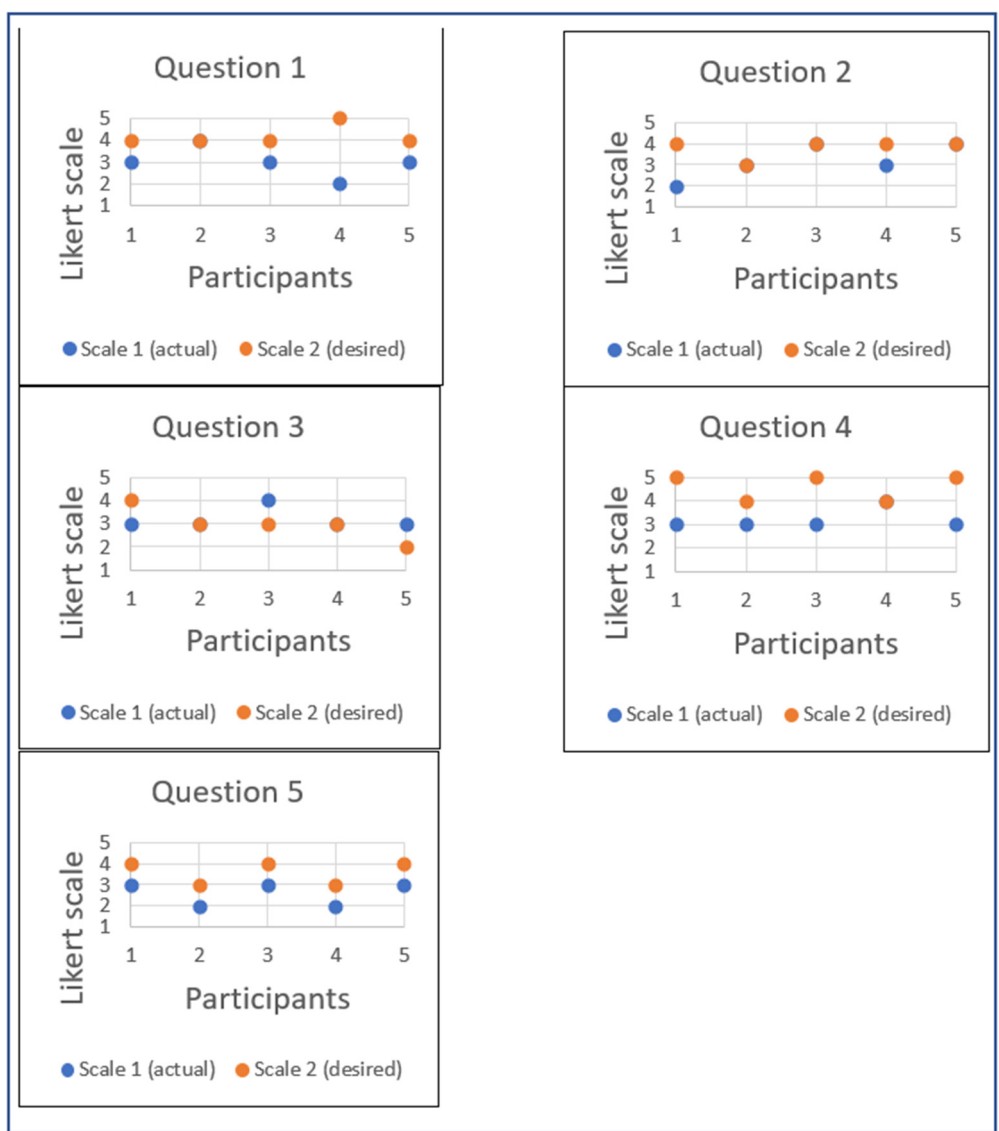

**Figure A1.** Scatter plots of the five questions asked to participants, described in Table 2.

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
