# Peer review of "Participatory Design of Sonification Development for Learning about Molecular Structures in Virtual Reality"

_mti, doi:10.3390/mti6100089_

Round 1

Reviewer 1 Report

The present study offers a slightly above-average literature review but could be more up-to-date. The research topic is exciting and original, as is the research design.

However, the submitted text does not meet the requirements for a research paper mainly for the following reasons:

- Completely non-transparent presentation of results. It is not clear what exactly was measured.

- Lengthy illustrative lists lead to an unhelpful lengthening of the text.

- The research sample is too small.

- The conclusions of the study are not linked to the results.

- Lack of formalized treatment and analysis of results.

The entire manuscript comes across as a preparation or outline written to provide feedback to the authors more than a final paper intended for publication. I also recommend paying attention to the formal (uniform referencing, formatting of figures and tables, etc.) aspects of the manuscript.

Author Response

The present study offers a slightly above-average literature review but could be more up-to-date. The research topic is exciting and original, as is the research design. 

However, the submitted text does not meet the requirements for a research paper mainly for the following reasons:

- Completely non-transparent presentation of results. It is not clear what exactly was measured.

We measured the application of PD in developing molecular sonification and the usability of the sonification.

We wrote this paragraph in the conclusions:

“The positive usability testing results provided an important insight on the usefulness and ease of use of the sonifications. These results support the importance of HCI in the sonification development process.

Our findings answered our research question, confirming that the participatory design methodology effectively supports the design of non-speech sounds to be used in scientific sonification for supporting learning of molecular structure. This is in line with the results obtained from the end-user computing development involvement questionnaire that we applied to the PD.”

The usability testing data can be downloaded from here:

 Data Availability Statement: Data supporting reported usability testing results can be found at: http://people.algomau.ca/garcia/PD_sonification_SUS_questionnaire_results.xlsx

- Lengthy illustrative lists lead to an unhelpful lengthening of the text.

We have reduced the length of the Introduction section and improved some paragraphs.

- The research sample is too small.

We added this explanation:

In a small-scale usability test such as the one we carried out, it has been calculated that 15 participants could uncover up to 100% of the usability problems of a software application [62]. More participants will be needed in further studies to get conclusive results in terms of educational gains of our developed sonification

- The conclusions of the study are not linked to the results.

The results of the usability testing were discussed at the beginning of the Discussion section.

We added the following in the Conclusions section: The positive usability testing results provided an important insight on the usefulness and ease of use of the sonifications. These results support the importance of HCI in the sonification development process.

We also wrote the following:

“Our findings answered our research question, confirming that the participatory design methodology effectively supports the design of non-speech sounds to be used in scientific sonification for supporting learning of molecular structure. This is in line with the results obtained from the end-user computing development involvement questionnaire that we applied to the PD.”

- Lack of formalized treatment and analysis of results.

Both quantitative results (the usability test results) and qualitative results were analyzed and discussed. We added this paragraph:

“We conducted a thematic analysis [61,62] on the qualitative data obtained from the focus group that we ran with the student participants. We identified four main themes related to sound parameters that can be used for making molecular sonifications:

  • Use melodic sounds
  • Use natural sounds (sound effects)
  • Change musical notes’ pitch
  • Change motifs’ length (a motif is a short melodic piece containing a succession of musical notes)

Regarding the comment on using five different musical instruments, all the students agreed that those instruments could be useful for representing the different types of hydrophobicity. According to the thematic analysis, some students suggested to slightly change the notes’ tempo and pitch from the sonifications. This may help future users discriminate notes among all the sonifications.”

The entire manuscript comes across as a preparation or outline written to provide feedback to the authors more than a final paper intended for publication. I also recommend paying attention to the formal (uniform referencing, formatting of figures and tables, etc.) aspects of the manuscript.

We have revised the formatting of figures and tables. We fixed their captions.

We sincerely thank the reviewer for his/her valuable feedback.

Reviewer 2 Report

This manuscript explores the application of the participatory design approach in the molecular sonification development in VR environments, a timely and important topic. The paper is well-written, following a sound methodology. Results presentation should be improved as explained later. More specific, the following issues have been identified:

L27: The introduction is thorough and easy to read. It presents the study’s main concept and research fields. However, what is missing is the contemporary developments in the intersection of audio & sonification with VR e.g. [1,2].

Additionally, the background information could be enriched with participatory approaches of VR systems design in educational settings e.g. [3,4].

L167: The reference (Temple, 2017) is missing.

L412: Check the syntax and grammar of the sentence. Is there a word missing after ‘very’, e.g. large?

L514: No results are presented in Table 1, please revise.

Finally, I encourage authors to add a short video abstract that illustrates the use of sounds in a VR environment.

1.          Serafin, S.; Geronazzo, M.; Erkut, C.; Nilsson, N.C.; Nordahl, R. Sonic Interactions in Virtual Reality: State of the Art, Current Challenges, and Future Directions. IEEE Comput Graph Appl 2018, 38, 31–43, doi:10.1109/MCG.2018.193142628.

2.          Tinoca, L.; Piedade, J.; Santos, S.; Pedro, A.; Gomes, S. Design-Based Research in the Educational Field: A Systematic Literature Review. Educ Sci (Basel) 2022, 12, 410, doi:10.3390/educsci12060410.

Author Response

This manuscript explores the application of the participatory design approach in the molecular sonification development in VR environments, a timely and important topic. The paper is well-written, following a sound methodology. Results presentation should be improved as explained later. More specific, the following issues have been identified:

L27: The introduction is thorough and easy to read. It presents the study’s main concept and research fields. However, what is missing is the contemporary developments in the intersection of audio & sonification with VR e.g. [1,2].

We added the following:

“Spatial audio and sonification in virtual reality (VR) have been recently researched due to computer hardware and software advancements, enhancing user immersion and positional sound perception [24]. Sonification in VR in turn supports comprehension of three-dimensional molecular structure models displayed in VR by offloading information from the human visual channel, as well as making visual information more salient in a 3D virtual environment [25–27] among other applications.”

Additionally, the background information could be enriched with participatory approaches of VR systems design in educational settings e.g. [3,4].

We have added the following:

“PD has been recently applied in the development of educational virtual reality systems. [39] report a PD-based methodology where students are central in educational VR design, allowing them to participate actively in its development. PD can also be applied to developing educational VR for science teaching. [40] created an educational VR environment for teaching about ocean acidification, developed in conjunction with marine educators. By participating in the educational VR development, marine educators found that 3D visualization, empowerment, and perspective-taking may support teaching ocean acidification.”

 We cited the references that you suggested, thank you.

L167: The reference (Temple, 2017) is missing.

We fixed that.

L412: Check the syntax and grammar of the sentence. Is there a word missing after ‘very’, e.g. large?

We revised all the sentences that contain the word ‘very,’ but we could not find any syntax or grammar issues in the text.

L514: No results are presented in Table 1, please revise.

We fixed that. The table number was wrong. We changed it for: “Table 2 indicates that although…”

Finally, I encourage authors to add a short video abstract that illustrates the use of sounds in a VR environment.

 We added the following:

“A video abstract that illustrates the use of the sonification in the VR environment running on Sketchfab is found in: http://people.algomau.ca/garcia/PD_video.mp4.”

We sincerely thank you for your valuable feedback.

Reviewer 3 Report

Strengths of the paper:

The authors present a timely and interesting research regarding participatory design of sonification development in order to learn molecular structures using virtual reality technologies. 

The methodology is well presented, and the results are shown in detail, and in general, are very interesting.

The language of the paper is acceptable, and easy to read, although there are some typos in the text. These does not influence readability, but if possible, should be corrected.

The number of references is adequate.

Weaknesses of the paper:

While not a weakness per se, the paper is quite long. Maybe the introductory section could be kept a bit more concise by deleting a few sentences. However, the cited references should be kept in the paper.

All in all:

Overall, this is a fine paper and the reviewer believes that it does not have larger faults.

The reviewer encourages the authors to continue this research.

Author Response

The authors present a timely and interesting research regarding participatory design of sonification development in order to learn molecular structures using virtual reality technologies. 

The methodology is well presented, and the results are shown in detail, and in general, are very interesting.

The language of the paper is acceptable, and easy to read, although there are some typos in the text. These does not influence readability, but if possible, should be corrected.

We have proofread the text. Also, there were some words that Word editor marked as typos such as PyMOL, but those are proper nouns. Thank you.

The number of references is adequate.

Weaknesses of the paper:

While not a weakness per se, the paper is quite long. Maybe the introductory section could be kept a bit more concise by deleting a few sentences. However, the cited references should be kept in the paper.

We took away these paragraphs from the Introduction section with is corresponding adaptation in the text to make the Introduction section more concise, keeping the references in the paper:

“In addition, misconceptions occur in student understanding of chemical bonding [2,3], basic yet abstract concept related to bond lengths and energies [4]. This is necessary to learn the nature of chemical reactions and the nature of physio-chemical properties of molecules. In addition, students have difficulty in comprehending molecular scale, since in real life it is in the range of nanometers or picometers, happening at the smallest microscopic scale. A typical organic amino acid molecule (the building blocks of proteins and larger molecules) has a size in the order of 0.3 nanometers [5] Thus, the scale and three-dimensional shape of molecules make the amino acids’ structural conformation more complicated to understand and conceptualize [6,7].”

“Although students usually learn how to deal with molecular geometries using different formats and molecular representations (e.g., Fisher projections, Lewis structures, etc.), they still require the ability to construct spatial reasoning and understand the three-dimensional arrangement of molecular structures”

“There is a particular human auditory capability so-called “cocktail party effect”, where the human ear can focus on a sound of interest from a numerous or complex background of other sounds played at the same time”

“Both computer-based and physical molecular models have been used in educational settings, but molecular model visualization alone has not been enough to support comprehension of some abstract key molecular concepts and properties”

“For example, a group of piano notes could represent (be mapped to) the number of bonds between two molecules and could be played when a student makes a click with the mouse on the graphical bond shown. Similarly, a sound of a waterfall could represent the molecular property of hydrophilicity (the molecule is attracted to water). Thus, the semantic meaning of the sounds used in the sonification could help the user remember the mapping between the sounds and the molecular information they represent.”

“An early application of sonification is the Geiger counter, an instrument that alerts users of invisible radiation levels by making a clicking sound, changing its tempo in response to the amount of radiation detected”

“A sonification can be interactive, where the user interacts with, select, or manipulate sonified data and perceives its sonification in real time. A non-interactive sonification technique is called audification, where data series are directly converted to samples of a sound signal and played back without interruption”

“According to [15], the main functions of information representation with sound include:

  • alarms, warnings and alerts,
  • monitoring processes, messages, or status,
  • applications in arts, sports and entertainment,
  • disambiguate or enhance visual data,
  • data exploration.”

“In this paper, we are interested in the last two functions, where we create molecular data sonification for its further exploration and enhancing visual analysis of molecular structure to be done by biology and chemistry students.”

“An example of an auditory icon can be a sound effect that can be heard when a user sends a document to the wastebasket icon in Apple’s Macintosh operating system, working as auditory feedback, just like the sound of paper being torn in real life.”

[34] define PD as follows:

“A process of investigating, understanding, reflecting upon, establishing, developing, and supporting mutual learning between multiple participants in collective ‘reflection-in-action’. The participants typically undertake the two principal roles of users and designers where the designers strive to learn the realities of the users’ situation while the users strive to articulate their desired aims and learn appropriate technological means to obtain them” (p. 2).

All in all:

Overall, this is a fine paper and the reviewer believes that it does not have larger faults.

The reviewer encourages the authors to continue this research.

We sincerely thank you for your valuable feedback.

Round 2

Reviewer 1 Report

Thank you for the revised manuscript. I think many interventions have helped the text. I think the topic of sonification is still relatively marginal, and it is essential that the authors address it. Having made the edits, I still suggest the following changes:

1. Table 3 shows the only presentation of qualitative research. From the authors' answers, it is clear that this is not a mixed design study but qualitative research. Therefore, more work needs to be done with the respondents' answers. In my opinion, they should form the core of the results chapter.

2. Table 2 presents the quantitative part of the research, which leads me to believe that: 1) The results should be presented in a graph, including a scatter plot. The table can be an excellent addition to the appendix. However, a promising treatment should be offered here, as well as an interpretation of the data, why such a sample is sufficient, how the variance varies across questions, and what it means.

These two observations are methodological and, at the same time, will be most evident in the results. I believe that if the authors incorporate them, a scientifically acceptable manuscript with significant potential impact added by the authors as non-public data should be included in the Figures and their interpretation directly in the manuscript.

I still recommend going through the resolution of all the Figures and adding them in print quality.

Author Response

Thank you for revising the manuscript. Here are our additions to the manuscript regarding Tables 2 and 3:

For Table 3 (which is now Table 4), we added this qualitative analysis:

"It is interesting to note that the musical parameters and features mentioned by the students in the focus group were the following:

  • melody
  • pitch
  • timbre
  • chord
  • length
  • scale
  • motif

Those musical parameters and features have been used in sonification research elsewhere, e.g., [13,71].

It is also worth mentioning that results of the qualitative also show that the students did not mention the sound parameter of loudness. Loudness is a subjective measure of how intensely human ears perceive a sound and can be used in three-dimensional positional sound simulation. This may be useful for identifying the position of the sonification of a molecular piece in a molecular structural model in virtual reality [25]. In addition, loudness has been also used in molecular sonification for identifying amino acids from molecular structural models [71]."

Regarding Table 2, we added Figure 6 showing the questionnaire results as a graph. We also added Appendix 1 showing all the  questions' results as scatter plots. We added Table 3 showing the standard deviations of the results and the following description:

"Table 3 shows that question number 2 has the highest standard deviation, where perhaps participants were more hesitant to determine the end user’s information needed (e.g., what kind of musical sounds were needed) for designing the sonifications.

We carried out a qualitative study by running a focus group with the students participating in the sonification development, asking what kind of musical sounds could be used in our proposed molecular sonification system. The focus group coordinator (the first author) asked the students to focus on possible auditory parameters (e.g., timbre, tempo, musical scales, melody, loudness, pitch, length, etc.) as well as other musical features such as motifs and chords, to be used in the sonifications. The students with musical training were asked to explain the musical parameters and features to the biology students, which did not have musical training. This way, all the students knew their meaning before making comments on the sonification in the focus group. Table 4 shows participants’ relevant comments on the musical sounds to be used in the sonification. Their qualitative feedback is valuable and will be considered for developing further sonification prototypes."

Regarding the figure resolution, they look fine on our side. This is something that we will discuss with the editor if we need to improve them. Thank you.

Thank you again for your time taken in the manuscript revision.

Round 3

Reviewer 1 Report

Thank you for the modifications. However, I would consider revising the quality of the graphs, which have improved the text considerably, but at the same time, their aesthetic quality is now poor.